# The Genetic Architecture of Vascular Anomalies: Current Data and Future Therapeutic Perspectives Correlated with Molecular Mechanisms

**DOI:** 10.3390/ijms232012199

**Published:** 2022-10-13

**Authors:** Lăcrămioara Ionela Butnariu, Eusebiu Vlad Gorduza, Laura Florea, Elena Țarcă, Ștefana Maria Moisă, Laura Mihaela Trandafir, Simona Stoleriu, Minerva Codruța Bădescu, Alina-Costina Luca, Setalia Popa, Iulian Radu, Elena Cojocaru

**Affiliations:** 1Department of Medical Genetics, Faculty of Medicine, “Grigore T. Popa” University of Medicine and Pharmacy, 700115 Iasi, Romania; 2Department of Nefrology–Internal Medicine, Faculty of Medicine, “Grigore T. Popa” University of Medicine and Pharmacy, 700115 Iasi, Romania; 3Department of Surgery II—Pediatric Surgery, “Grigore T. Popa” University of Medicine and Pharmacy, 700115 Iaşi, Romania; 4Department of Mother and Child Medicine-Pediatrics, “Grigore T. Popa” University of Medicine and Pharmacy, 700115 Iaşi, Romania; 5Faculty of Dental Medicine, Odontology-Periodontology, Fixed Prosthesis Department, “Grigore T. Popa” University of Medicine and Pharmacy, 700115 Iasi, Romania; 6Department of Internal Medicine, “Grigore T. Popa” University of Medicine and Pharmacy, 16 University Street, 700115 Iasi, Romania; 7III Internal Medicine Clinic, “St. Spiridon” County Emergency Clinical Hospital, 1 Independence Boulevard, 700111 Iasi, Romania; 8Department of Surgery, Regional Institute of Oncology, I-st Surgical Oncology, “Grigore T. Popa” University of Medicine and Pharmacy, 700483 Iasi, Romania; 9Department of Morphofunctional Sciences I—Pathology, “Grigore T. Popa” University of Medicine and Pharmacy, 700115 Iasi, Romania

**Keywords:** vascular anomalies, signaling pathway, somatic mutation, germline mutation, second-hit mutation

## Abstract

Vascular anomalies (VAs) are morphogenesis defects of the vascular system (arteries, capillaries, veins, lymphatic vessels) singularly or in complex combinations, sometimes with a severe impact on the quality of life. The progress made in recent years with the identification of the key molecular pathways (PI3K/AKT/mTOR and RAS/BRAF/MAPK/ERK) and the gene mutations that lead to the appearance of VAs has allowed the deciphering of their complex genetic architecture. Understanding these mechanisms is critical both for the correct definition of the phenotype and classification of VAs, as well as for the initiation of an optimal therapy and the development of new targeted therapies. The purpose of this review is to present in synthesis the current data related to the genetic factors involved in the etiology of VAs, as well as the possible directions for future research. We analyzed the data from the literature related to VAs, using databases (Google Scholar, PubMed, MEDLINE, OMIM, MedGen, Orphanet) and ClinicalTrials.gov. The obtained results revealed that the phenotypic variability of VAs is correlated with genetic heterogeneity. The identification of new genetic factors and the molecular mechanisms in which they intervene, will allow the development of modern therapies that act targeted as a personalized therapy. We emphasize the importance of the geneticist in the diagnosis and treatment of VAs, as part of a multidisciplinary team involved in the management of VAs.

## 1. Introduction

Vascular anomalies (VAs) represent a heterogeneous group of blood vessel anomalies: capillaries, lymphatics, arteries, veins, or a spectrum of mixed anomalies, the severity of which varies from a simple birthmark to severe, life-threatening entities [1]. The estimated prevalence of VAs is 4.5%, and most appear sporadically and isolated, but there are also syndromic forms of the disease [1]. In 1982, Mulliken and Glowacki described the first clinical classification of VAs that also considered the histological characteristics of vascular endothelial cells (ECs). This classification, later adopted by the International Society for the Study of Vascular Anomalies in 1996, classifies and differentiates VAs into two types: vascular proliferative tumors and vascular malformations [2,3]. The distinction between the two entities is based on histopathological assessment of increased ECs turnover. Vascular tumors (most of which are hemangiomas) are histologically characterized by a high turnover of ECs. They represent the neoplastic growth of vascular ECs and usually have an evolution that includes a proliferative phase, a period of plateau or stability, followed by spontaneous regression. Vascular malformations are anomalies of dysmorphogenesis composed of vascular channels with an abnormal structure, lined by ECs, without their proliferation and a normal turnover of vascular ECs. They are congenital, but sometimes they are not noticed immediately at birth. They do not regress, and grow proportionally with the development of the individual (Table 1) [2].

Most of the VAs are present at birth or in the neonatal period and can develop later, but they can also appear later during the life of affected individuals. Etiologically, VAs are caused by inherited germline mutations or somatic mutations [3,4,5]. According to Knudson’s two-hit hypothesis, the second loss-of-function (LOF) mutation in somatic cells (in which a first mutation, inherited or acquired, is already present) will cause the loss of heterozygosity, the consequence being the LOF of the encoded protein in the affected cells and the appearance of VAs [3]. Most isolated VAs are caused by somatic gain-of-function (GOF) mutations of genes involved in angiogenesis, lymphangiogenesis, vascular cell proliferation and apoptosis, some of which are also detected in certain types of cancer [4].

The extreme phenotypic variability of VAs is correlated with the complex molecular mechanisms that involve numerous genes encoding molecules that intervene in different signaling pathways at the level of vascular cells: RAS (rat sarcoma)/RAF (rapidly accelerated fibrosarcoma)/MAPK (mitogen-activated protein kinase kinase)/ERK (extracellular signal-regulated kinase); angiopoietin/TIE2 (angiopoietin-1 receptor), PI3K (phosphoinositide 3-kinase)/AKT (protein kinase B)/mTOR (mammalian target of Rapamycin); TGFB (transforming growth factor beta) signaling, and the G protein–coupled receptor signaling molecules (GNA [G protein subunit alpha] Q/GNA11/GNA14) Figure 1) [4,5].

The aim of our paper is to provide an in-depth analysis of the data available in the literature on the role of genetic factors involved in the etiology of VAs. We focused on studying the complex molecular mechanisms involving numerous genetic factors that encode molecules that intervene in different signaling pathways. We highlighted the fact that both the genetic heterogeneity and the phenotypic overlaps between certain entities can create diagnostic difficulties, and the identification of new genetic factors could contribute to the development of new innovative therapies that act at the level of specific targets. In addition, the concept of a two-hit mechanism (Knudson’s two-hit hypothesis) could explain the incomplete penetrance and variable expressivity present in inherited multifocal venous anomalies that follow a monogenic autosomal dominant pattern of transmission, while sporadic cases whose etiology is not fully elucidated are caused by of somatic mutations of the same genes.

The data synthesized and presented in this review was obtained by examining the literature (Google Scholar, PubMed, MEDLINE, OMIM, MedGen, Orphanet databases, and ClinicalTrials.gov) and using the following keywords: vascular anomalies, vascular malformations, vascular tumors, signaling pathways, germline and somatic mutations, and Knudson’s two-hit mutation hypothesis (Table 2).

## 2. Vascular Anomalies: RAS/RAF/MAPK/ERK Signaling Pathways (RASopathies)

The main function of the RAS/RAF/MAPK/ERK signaling pathway, also called the “proliferation pathway”, is to transduce signals from the extracellular milieu to the cell nucleus where specific genes are activated for cell cycle regulation, proliferation, and cell migration. The mutations of these genes determine a group of diseases, called RASopathies, which associate with VAs [4,76].

Numerous growth factors, cytokines, and G protein–coupled receptor ligands induce signaling in the RAS/RAF/MAPK/ERK pathway, activating RAS by replacing GDP (Guanosine diphosphate) with GTP (Guanosine-5’-triphosphate). After binding to RAS, RAF exhibits serine/threonine-protein kinase activity and activates MAPK, which will subsequently activate ERK by phosphorylation [4].

The RAS/RAF/MAPK/ERK signaling pathway is also involved in cell cycle regulation, cell damage repair, integrin signaling, and can stimulate angiogenesis by altering the expression of genes directly involved in the formation of new blood vessels. Gene mutations involved in the RAS/RAF/MAPK/ERK pathway are correlated with tumorigenesis, the *RAS* gene being an oncogene whose mutations are frequently detected in human cancers [76,77].

### 2.1. Venous Malformations (VMs) and RAS/RAF/MAPK/ERK Signaling Pathway

Venous malformations (VMs) are slow-flow vascular lesions, caused by a defect in vascular morphogenesis during early embryonic life (weeks 4–10 of gestation), characterized by the cluster of dilated venous channels associated with a thin or absent vascular wall [8].

#### 2.1.1. Verrucous Venous Malformation (VVM)

Verrucous Venous Malformation (VVM) also called Formerly Verrucous Hemangioma, is a non-hereditary venous malformation, caused by activating somatic mutations (ASM) of the *MAP3K3* (Mitogen-Activated Protein Kinase Kinase Kinase 3) gene (OMIM 602539), located on chromosome 17q23.3 [6]. *MAP3K3* is involved in both the ERK and the AKT/mTOR signaling pathways. VVMs are clinically manifested by cutaneous capillary venous malformations (CVMs) present as well-demarcated purpuric linear plaques covered by a hyperkeratotic dermis, which can reach sizes of several centimeters. VVMs can be present at birth, or appear early in childhood, most frequently being located at the level of the lower limbs. In young patients, the lesions have a red-blue appearance, are soft, but become hyperkeratotic over time [7,8].

#### 2.1.2. Cavernous Cerebral Malformation (CCM)

Cerebral cavernous venous malformations (CCMs), also known as cavernous hemangiomas or cavernomas, are slow-flow VMs consisting of a “mulberry-like” cluster of hyalinized dilated thin-walled capillaries, without intervening normal brain tissue. CCMs affects up to 0.5% of the general population [8]. Due to recurrent microhemorrhages and thrombosis, they are usually surrounded by hemosiderin deposits and gliosis. The supratentorial location is most common, although lesions may also occur in the basal ganglia, brainstem, cerebellum, and spinal column. Occasionally, CCMs may be associated with a developmental venous anomaly (VAV), in which case they are known as a mixed vascular malformation [37].

Most cases appear sporadically and are represented by single lesions, which become symptomatic around the age of 40–60 years, but there can also be multiple lesions, often having a familial characteristic. Most of the CCMs remain asymptomatic throughout life, being detected incidentally following neuroimaging investigations. CCMs causes symptoms ranging from headaches, epilepsy, focal neurological deficit, to extensive life-threatening cerebral hemorrhages [4,37].

Although the vascular lesions, characteristics of CCMs, are frequently located in the central nervous system (CNS), they can often associate lesions in the skin, retina, kidneys, and liver [4,6].

Cutaneous vascular malformations are present in approximately 9% of patients with CCMs, three distinct phenotypes being described as: hyperkeratotic cutaneous capillary -venous malformations (HCCVM) (39%), capillary malformations (CMs) (34%), and venous malformations (VMs) [6,8]. 

Hyperkeratotic cutaneous capillary—venous malformations (HCCVM, OMIM 116860) are rare cutaneous lesions that occur in a group of patients with CCMs [6]. The cutaneous lesions are congenital and present as thick, irregular, black or purpuric plaques, localized especially on the limbs [4,6,9]. Capillary malformations (CMs) are usually congenital and appear as a port-wine stain or the so-called “punctate” capillary malformation. Venous malformations (VMs) in patients with CCMs can appear as single nodules (often located in one limb) or multiple nodules of variable size [8].

Familial forms are determined by mutations transmitted in an autosomal dominant manner with incomplete penetrance, involving the *CCM1 (KRIT1)* gene, located on chromosome 7q21–22 [10, the *CCM2* (*MGC4607* or malcavernin gene) located on chromosome 7p13–15 [11], and the *PDCD10 (CCM3)* gene located on chromosome 3q25.2–27 [12]. The first mutations identified were a LOF mutations in the *KRIT1* gene, which encodes the protein KRIT1 (KREV1 interaction trapped 1), an evolutionarily conserved Ras-family GTPase. So far, it is not precisely known the role of *KRIT1* in the formation of cerebral capillaries and veins. The only thing known is that *KRIT1* interacts with KREV1/RAP1A, a GTPase of the *RAS* family. The *KRIT1/CCM2/CCM3* complex is known to inhibit *MAP3K3*. Loss-of-function of *KRIT1* and therefore loss of the entire CCM complex leads to activation of MAP3K3 signaling [8].

All the mutations identified in the CCM genes are LOF mutations, which cause a deficiency of the encoded proteins, causing molecular disorganization and dysfunction of endothelial junctions, affecting the maintenance of the integrity of the vascular barrier. *KRIT1* (CCM1) mutations were detected most frequently in patients with HCCVM [8,10]. Recently, somatic mutations of the *MAP3K3, PIK3CA, MAP2K7* genes have been identified in CCM lesions, which would suggest that in the case of CCMs there may also be somatic mosaicism [13].

### 2.2. Capillary Malformations (CMs) and RAS/RAF/MAPK/ERK Signaling Pathway

#### 2.2.1. Capillary Malformations (CMs)

Capillary malformations (CMs) are detected in approximately 0.1–2% of newborns [78] and can be isolated (cutaneous CMs) or be part of a complex syndrome. CMs are usually sporadic and appear as flat, red to purple lesions named port-wine stain (nevus flammeus). The etiology of CMs is represented by ASM in the *GNAQ* gene, located on chromosome 9q21.2 [6,14,15].

Sturge-Weber syndrome (SWS) (OMIM 185300) [6] is a neurocutaneous disorder characterized by capillary malformation (port-wine stains), and choroidal and leptomeningeal vascular malformations most often involving the occipital and posterior parietal lobes [6]. The most common symptoms and signs are facial CM (port-wine stains) typically on the forehead and upper eyelid in the distribution of the 1st and/or 2nd division of the trigeminal nerve, seizures, and glaucoma [6]. Until now, whole genome sequencing (WGS) studies have highlighted the presence of the recurrent somatic mutation c.548G>A (p.R183Q) in the *GNAQ* (G-alpha q) gene, both in SWS patients and in non -syndromic port-wine stain lesions. The *GNAQ* p.Arg183Gln mutation that causes the loss of arginine in *GNAQ* with the reduction of hydrogen bonds between G(q) and GDP is most frequently detected in patients with CMs [14].

In their study, Shirlley et al. [14] identified a nonsynonymous single-nucleotide variant (c.548G→A, p.Arg183Gln) in the *GNAQ* gene in 88% of patients with SWS and in 92% of patients with non-syndromic CMs (port-wine stains) [14]. They did not identify the respective mutation either in the case of four unrelated patients with cerebrovascular malformation, or in the 6 patients from the control group [14]. The somatic substitutions in *GNAQ* p.Gln209Leu and *GNAQ* p.Arg183Gln were detected in patients with uveal melanoma. The most common is the *GNAQ* p.Gln209Leu mutation which has been shown to overactivate the mitogen-activated protein kinase (MAPK) pathway. The *GNAQ* p.Arg183Gln mutation has a gain-of-function effect that activates downstream signaling pathways. However, the effect of *GNAQ* p.Arg183Gln in MAPK signal transduction appears to be weaker in terms of activation of downstream effectors than the effect of the more frequently detected *GNAQ* p.Gln209Leu substitution in uveal melanoma tissue [14]. Galeffi et al. [15] identified the presence of the *GNAQ* p.R183Q mutation in most of the patients with SWS that were analyzed, and in one patient, a new *GNAQ* Q209R mutation was identified [15].

#### 2.2.2. Capillary Malformations (CM)—Arteriovenous Malformation (AVM)

Cutaneous vascular lesions specific to patients with Capillary Malformation (CM)—Arteriovenous Malformation (AVM) (CM-AVM) are small, multifocal, disseminated, round/oval shaped, and pink to red colored macules or papules surrounded by a pale halo on Doppler ultrasound [16]. They are present at birth, but can also appear later in life. CM-AVMs are frequently associated with high-flow arteriovenous malformations (AVMs) or arteriovenous fistulas (AVFs) located in muscles, skin, and other tissues (intracranial, intraspinal), aneurysmal malformation of the vein of Galen or Parkers Weber syndrome (PWS) [16,17,18].

Mutations of the *RASA1* gene (RASp21 protein activator 1; OMIM 139150) [6] (located on the chromosome 5q14.3) and transmitted in an autosomal dominant manner, are responsible for different forms of VMs (CMs, AVMs, AVFs), as single manifestations, or in complex combinations, as in the case of PWS [16,18]. Initially, two loci for inherited CMs were identified on chromosome 5 (5q14-21 and 5q13-22), and later, the *RASA1* (RAS P21 Protein Activator 1) gene was identified as a candidate gene for atypical CMs with AVMs and AVFs, and sporadically, in PWS [16,18].

The spectrum of capillary malformation—arteriovenous malformation syndrome 1 (CM-AVM1) includes all cases of VMs associated with *RASA1* mutations. Later, other studies reported the presence of *RASA1* mutation in atypical CMs as well as cases with aneurysmal malformation of the vein of Galen (VOGM) and Hereditary hemorrhagic telangiectasia (HHT) [18].

CM-AVM1 syndrome follows an autosomal dominant inheritance pattern with incomplete penetrance and variable expressivity (this aspect may suggest the involvement of a two-hit mechanism). The *RASA1* gene encodes a GTPase-activating protein (p120-RasGAP) and is a negative regulator of RAS/MAPK and MAPK/ERK pathways, transforming Ras protein into its inactive form, variations in p120-RasGAP protein level, having an impact on angiogenesis [19]. The active GTP form of Ras interacts with the Raf protein which is responsible for the phosphorylation of proteins involved in cell growth, proliferation, and differentiation. After activation of the receptor tyrosine kinase, p120-RasGAP is recruited to the cell membrane where it inhibits the RAS/MAPK/ERK signaling pathway and regulates cell growth, differentiation, and proliferation. The p120-RasGAP protein interacts with p190RhoGAP or FAK (focal adhesion kinase), both of which have a role in the movement of the vascular endothelium [19].

Unlike CM-AVM1, CM-AVM2 is caused by LOF mutations of the *EPHB4* gene, located on chromosome 7q22.1 [20]. The *EPHB4* gene encodes the protein EPHB4 (Ephrin type-B receptor 4), a transmembrane tyrosine kinase receptor that binds to ephrin-B2 and plays an essential role in vasculogenesis [6]. EPHB4 inhibits the RAS/MAPK/ERK pathway by interacting with p120-RasGAP. This inhibitory effect is lost in CM-AVM2 leading to a constitutive activation of RAS/MAPK/ERK signaling [4,20].

### 2.3. Arteriovenous Malformations and RAS/RAF/MAPK/ERK Signaling Pathway

Arteriovenous Malformations (AVMs) can be located in any organ in the body, including visceral or peripheral structures, and in the CNS. AVMs often destroy adjacent structures during their development/growth, being the most aggressive vascular malformations. Most of the time, it is not possible to completely remove the lesion by embolization or surgical intervention, and the remaining lesions cause the worsening of the AVM [4,19].

In their study, Couto et al. [21] identified the presence of a ASM mutation in the *MAP2K1* (Mitogen-activated protein kinase kinase 1) gene, located on chromosome 15q22.31) [6] in seven of the ten analyzed patients, who presented peripheral or extracranial AVMs [21]. The ASM mutations in the *MAP2K1* gene (which encodes MAP-extracellular signal-regulated kinase 1, MEK1), activate the RAS/MAPK signaling pathway that controls numerous cellular and developmental processes. Somatic mutations in the *KRAS* (Kirsten rat sarcoma virus) gene, located on chromosome 12p12.1 (OMIM 190070) [6] identified by Couto et al. [21] have also been identified in various neoplasms (including melanoma, lung cancer, and hematopoietic malignancies) and have been shown to constitutively increase MEK1 activity [21]. Somatic mutations that affect the proteins that intervene upstream of MEK1, are detected both in different types of vascular malformations and in different types of cancer [21,22].

Nikolaev et al. [23] detected two ASM mutation (c.35G→A, p.Gly12Asp and c.35G→T, p.Gly12Val), in the *KRAS* gene, in the case of 45 of the 72 patients with cerebral AVMs [23]. The authors demonstrated that the expression of the mutant *KRAS* gene in endothelial cells (ECs) in vitro induces an increase in ERK activity, and an increase in the expression of genes involved in angiogenesis and Notch signaling (Notch homologous protein of the neurogenic locus), as well as increase of cell migration capacity [23]. These processes were reversed by inhibiting MAPK (mitogen-activated protein kinase)–ERK signaling [23].

In another study, Al-Olabi et al. [24] identified mutations in *KRAS, BRAF* (proto-oncogene B-raf) (localized on chromosome 7q34) and *MAP2K1* genes, which supports the role of RAS/RAF/MAPK signaling in AVMs [24].

### 2.4. Lymphatic Malformations and RAS/RAF/MAPK/ERK Signaling Pathway

Gorham-Stout disease (GSD) (OMIM 123880), which is also known as “vanishing bone disease”, “disappearing bone disease”, or cystic angiomatosis of bone, is a rare disease of massive osteolysis associated with proliferation and overgrowth of lymphatic vessels [6]. GSD belongs to complex lymphatic malformations (LMs) [6] and may affect any bone in the body and can be monostotic or polyostotic. The ribs, spine, pelvis, skull, collarbone (clavicle), and jaw are frequently affected. Symptoms at presentation are dependent upon the location(s) of the disease; the most common symptom is localized pain accompanied by swelling of the affected area and functional impotence. The disease may be discovered after a pathological fracture determined by osteolysis and osteopenia. The etiology of GSD is not fully known, and the anatomopathological data indicate disorganized lymphangiogenesis [6,25].

Homayun-Sepehr et al. [26] identified an ASM mutation (p.G12V) in the *KRAS* gene in a patient with GSD [26].

Kaposiform lymphangiomatosis (KLA) is a rare, frequently aggressive, systemic disorder of the lymphatic vasculature, characterized by multifocal malformed lymphatic channels, frequently located in the thoracic cavity, but also involving the spleen or the skeleton. Specific clinical manifestations include pericardial and pleural effusions, cough, dyspnea, bleeding, and fractures secondary to bone involvement, with poor prognosis [27].

Barclay et al. [28] identified an ASM in the *NRAS* gene (c.182A>G, p.Q61R) in 10 of the 11 patients with KLA, suggesting that RAS signaling is important for the development of KLA. The *NRAS* gene located on chromosome 1p13.2 [6] encodes the N-Ras protein that is involved primarily in regulating cell division [28]. In addition, the activating *NRAS* p.Q61R variant is known as a hotspot variant, frequently identified in several types of human cancer, especially melanoma [28].

### 2.5. Vascular Tumors and RAS/RAF/MAPK/ERK Signaling Pathway

#### 2.5.1. Pyogenic Granuloma

Pyogenic granuloma (PG) (also called lobular capillary hemangioma) is an acquired benign vascular hyperplasia, with an etiology that is not fully known. PG occurs frequently in children and young adults, and manifests as a red papular lesion or a solitary, rapidly growing nodule, prone to bleeding from minor trauma and ulceration [4,29].

PG generally occurs in the skin and mucous membranes (face, trunk, oral cavity), but occasionally, it can be located in the gastrointestinal tract or larynx. PG can occur spontaneously or within capillary malformations (CMs). PG occurs frequently during pregnancy, being called “pregnancy tumor”, or associated with certain drugs. In secondary PGs, detected in some CMs, an ASM mutation in the *GNAQ* gene (p.Arg183Gln) has been described. Because the CM and concomitant PG have the same mutation, the hypothesis is that the PG originates from the underlying CM cells [29].

The pathogenesis of most sporadic PGs and PGs associated with port-wine stains (PWS) is not fully elucidated. Groesser et al. [29] analyzed 10 cases with PGs secondary to a PWS. The authors identified a *BRAF* mutation c.1799T>A (p.Val600Glu) in the case of 8 patients and a *NRAS* mutation c.182A>G (p.Gln61Arg) in one patient [29]. *GNAQ* mutation c.548G >A was identified in PGs and underlying PWS respectively, indicating that PGs originate from PWS cells [29]. In 25 patients with sporadic PGs, the authors identified the *BRAF* c.1799T>A mutation in 3 of the patients, a *BRAF* c.1391G>A mutation in one patient, and a *KRAS* c.37G>C mutation in one patient [29]. The authors concluded that the *BRAF* c.1799T>A gene mutation has a major role in the pathogenesis of PGs, especially of the secondary CMs, opening the way for deciphering the genetic basis of PGs [29].

Other studies have reported cases of PGs in which somatic mutations common to those detected in colon cancers have been identified, respectively mutations in the *BRAF* (p.Val600Glu or p.Gly464Glu), *KRAS* (p.Gly13Arg), *GNA14*, and *HRAS* (p.Q61R, p.E49K, p.Q61R and p.G13S) genes [29,30,31].

Relatively recently, cases of oral PGs were reported, in which no *BRAF, KRAS, HRAS, NRAS, GNA11,* or *GNA14* gene mutations were identified, suggesting that although oral PGs shows activation of the MAPK/ERK pathway, the major molecular events are not fully elucidated [32].

#### 2.5.2. Congenital Hemangioma

Congenital hemangioma (CH) is a rare vascular tumor that forms during intrauterine development. CH are different from common infantile hemangiomas (IH) that enlarge rapidly after birth and immunostain for the cell surface marker GLUT1.1. In contrast, CHs are negative for GLUT1.1, and postnatally, the tumor either rapidly involutes (rapidly involuting congenital hemangioma) (RICH) or regresses (non-involuting congenital hemangioma) (NICH) [33,34].

Starting from the premise that CHs are determined by somatic mutations, Ayturk et al. [33] performed mRNA sequencing from affected tissue in a patient with CH. The authors identified the mosaic missense mutations that alter glutamine at amino acid 209 (pGlu209Leu) (c.626A>T) in the *GNAQ* or *GNA11* genes in all tested samples. These mutations were different from those detected in capillary malformations (CMs). The authors then looked for the presence of mutations in the genomic DNA. Both *GNAQ* and *GNA11* pGlu209Leu missense variants identified in patients with CHs are also common mutations in uveal melanoma and have been shown to constitutively activate MAPK and/or YAP (Yes-Associated Protein) signaling [33].

## 3. Vascular Anomalies: PI3K/AKT/mTOR Signaling Pathways (Pikopathies)

The PI3K/AKT/mTOR pathway is an important intracellular signaling pathway in the regulation of the cell cycle, being directly linked to cell proliferation and cancer. Activation of PI3K (Phosphoinositide 3-kinases) phosphorylates and activates AKT (AKT serine/threonine kinase 1) localizing it to the plasma membrane. AKT also has downstream effects: CREB (CREB, cAMP response element-binding protein) activation, p27 inhibition, FOXO1 (Forkhead Box O1) localization in the cytoplasm, Phosphatidylinositol 3-phosphate (PtdIns-3ps) activation and mTOR activation that can affect the transcription of p70 or 4EBP1 (Eukaryotic translation initiation factor 4E-binding protein 1) [4,5]. Many activators of the PI3K/AKT signaling pathway are known, including EGF (epidermal growth factor), Shh (Sonic hedgehog protein), IGF-1 (Insulin-like growth factor 1), insulin, and CaM (Calmodulin) [5]. Insulin and Leptin recruit PI3K signaling for metabolic regulation, and PTEN, GSK3B and HB9F negatively regulate the PI3K/AKT pathway. VAs caused by signaling defects in the PI3K/AKT/mTOR pathway are called PIKopathies. Hereditary hemorrhagic telangiectasias (HHT) and various slow-flow malformations (especially venous and lymphatic malformations) are caused by activating mutations of the AKT/mTOR pathway [5].

### 3.1. Venous Malformations (VMs) and PI3K/AKT/mTOR Signaling Pathway

#### 3.1.1. Focal and Multifocal Venous Malformations

Venous Malformations (VMs) are blue, soft, and compressible lesions, and belong to slow-flow Vascular Malformations [4]. Histologically, VMs are developmental errors composed of dysmorphic, dilated venous channels lined by flattened endothelium that exhibit slow turnover. The lesions are always present from birth, most frequently as an isolated, singular manifestation, but they can have multifocal localizations [35]. They can manifest clinically in childhood or adulthood, but depending on their location, they can remain asymptomatic throughout life. VMs frequently involve the skin (face, limbs, trunk) and mucous membranes, but they can also be present at the level of internal organs, the skeleton, and skeletal muscle. The primary form of non-surgical therapeutic intervention for VMs is represented by sclerotherapy [35].

Activating somatic mutations (ASM) of the *TEK* gene (also called the *TIE2* gene), located on chromosome 9p21.2 [6] are the most frequently detected mutations, both in the majority of unifocal, sporadic VMs, as well as in inherited cutaneomucosal venous malformations (VMCMs). The *TEK* gene encodes a protein called TEK receptor tyrosine kinase, which is expressed almost exclusively in ECs in mice, rats, and humans. TEK acts as a cell surface receptor for angiopoietin (ANGPT). ANGPT1, ANGPT2, and ANGPT4 regulate angiogenesis, proliferation, migration, adhesion, and migration of ECs, but also the maintenance of vascular quiescence. TEK (TIE2) is activated by ANGPT1 (its ligand), whereas ANGPT2 is able to modulate TIE2 activity in a context-dependent manner. Ligand binding results in the activation of TEK (TIE2), which causes activation of the PI3K/AKT/mTOR signaling pathway [36,37].

More than 20 different mutations are described in the literature, which occur at the level of the intracellular domain of RKTs (Receptor tyrosine kinases) (kinase domain, kinase insert domain, or carboxyl-terminal tail) that determine amino acid substitutions or truncation of the inhibitory C-terminal loop [36,37].

*TEK* L914F is the most common mutation detected in sporadic VMs. The *TEK* R849W mutation is frequently detected in VMCMs, transmitted in an autosomal dominant manner with incomplete penetrance, and causes weak autophosphorylation of TEK (TIE2) when overexpressed in human umbilical vein ECs (HUVECs) [37].

Unlike the usual unifocal VMs, which is most often caused by the somatic *TEK* L914F mutation, the multifocal VMs are predominantly caused by double (cis) mutations (two somatic mutations in the same allele) [38]. The production of typical multifocal lesions involves a second-hit mutation in *TEK* gene. Most frequently, patients present a somatic mosaicism already having the first mutation (most frequently the *TEK* R915C mutation), subsequently producing the second mutation, frequently detected in the affected areas [8,38]. In the study of Nätynki et al. [38], the TIE2 protein/mRNA ratio was strongly decreased in the presence of both mutations of the *TEK* Y897F-R915L, compared to cells in which only one mutation was present [38].

#### 3.1.2. Blue Rubber Bleb Nevus Syndrome (BRBN)

Blue rubber bleb nevus syndrome (BRBN) (also called Bean syndrome) (OMIM, 112200) [6] is a rare condition that occurs sporadically, and is characterized by the presence of numerous cutaneous and visceral (especially gastrointestinal) VMs, often causing consumptive coagulopathy and chronic anemia [6]. Soblet et al. [39] identified the presence of somatic mutations in *TEK* gene in 15 of the 17 patients with BRBN [39]. In the same study, the authors identified somatic mutations in five out of six individuals with multifocal VMs (sporadic forms). *TEK* Y897F-R915L mutations have been identified mostly in cases with multifocal VMs, and less often in individuals with BRBN, for whom *TEK* T1105N-T1106P mutations are specific [39]. Both types of mutations cause TEK ligand-independent activation and increase ECs survival, invasion and colony formation when expressed in human umbilical vein (HUVEC) [39].

In vitro, it has been shown that all *TEK* (*TIE2*) mutations lead to ligand-independent hyperphosphorylation of the receptor and a permanent activation of the PI3K/AKT/mTOR signaling pathway, the effect being stronger in the presence of a double mutation, compared to the presence of a single mutation of the *TEK* gene [39,40].

VMs in which the *TEK* L914F mutation was identified showed a lower expression of platelet-derived growth factor beta (PDGFB) and α-Smooth muscle actin (α-SMA) than in normal veins. PDGFB, as an EC-secreted attractant, plays an essential role in pericyte recruitment. Mutations in the TEK/*TIE2* gene significantly impede the ability of ECs to produce PDGFB. Downregulation of PDGFB depends on AKT phosphorylation (FOXO1-Mediated Activation of AKT) [40]. Si et al. [40] showed that the *TEK* L9144F mutation acts through the AKT/FOXO1/PDGFB pathway to produce VMs. Based on this proven fact, AKT pathway inhibitory therapy could represent an effective therapeutic strategy in the case of VMs caused by the *TEK* L9144F mutation [40].

#### 3.1.3. Venous Anomalies and PIK3CA-Related Overgrowth Syndrome

Activating somatic mutations (ASM) in the *PIK3CA* (Phosphatidylinositol 4,5-Bisphosphate 3-Kinase Catalytic Subunit Alpha) gene located on chromosome 3q26.32 [6] were detected in a small number of VMs that appear sporadically [41].

Phosphoinositide-3-kinase (PIK3) is composed of an 85 kDa regulatory subunit and a 110 kDa catalytic subunit. The *PIK3CA* gene encodes the p110α catalytic subunit of PI3K, the downstream effector of *TEK* (*TIE2*), being considered an integral part of the PIK3 pathway. Mutations of both *PIK3CA* and *TEK* genes cause chronic activation of AKT. The *PIKA3* gene is an oncogene, whose hotspot mutations c.1624G>A (Glu542Lys), c.1633G>A (p.Glu545Lys,) in the helical domain and c.3140A>G (p.His1047Arg) in the kinase domain are frequently detected in different types of cancer, especially breast and cervical cancer [41,42,43].

*PIK3CA* and its interaction with the AKT and mTOR pathways are the subject of numerous research studies, and therapy that inhibits PI3K, even if it had limited efficacy as monotherapy, seems promising, in a combination therapy with other inhibitors (TKIs, MEKs, PARP, respectively aromatase inhibitors in breast cancer) [4].

*PIK3CA* mutations have also been detected in different PIK3CA-related overgrowth spectrum (PROS), in which VMs are associated with hypertrophy of soft and sometimes bone tissues [4].

PROS includes Megalencephaly-Capillary Malformation-Polymicrogyria syndrome (MCAP) (OMIM 602501) [6,42] and Congenital Lipomatous Overgrowth, Vascular Malformations, and Epidermal Nevi syndrome (CLOVES) (OMIM 612918) [6,44,45]. In both case, non-hotspot mutations of the *PIK3CA* gene are more frequently detected, compared to hotspot mutations, often detected in the form of somatic mosaicism [6,44,45].

Studies done over time have shown that the PI3K/AKT/mTOR pathway plays a crucial role in the occurrence of VMs, either due to *PIK3CA* or a *TEK* mutations [41,43].

### 3.2. Arteriovenous Malformations and PI3K/AKT/mTOR Signaling Pathway

#### 3.2.1. PTEN Hamartoma Tumor Syndrome (PHTS)

PTEN hamartoma tumor syndrome (PHTS) represents a spectrum that includes Cowden syndrome 1 (CWS1), Bannayan-Riley-Ruvalcaba syndrome (BRRS), PTEN-related Proteus syndrome (PS), and PTEN-related Proteus-like syndrome [6,46].

PHTS spectrum disorders are caused by LOF mutations in the *PTEN* (phosphatase and tensin homolog) gene located on chromosome 10q23.31 (OMIM 601728) [6] with an autosomal dominant transmission pattern [46]. The affected individuals present, along with the specific clinical manifestations (macrocephaly associated with multiple hamartromas), high-flow VAs with cutaneous or deep, subcutaneous localization, which are not pathognomonic for PHTS. Since *PTEN* gene inhibits the PI3K/AKT/mTOR pathway, mutations with the loss of its function leads to the activation of this signaling pathway, as well as predisposition to tumors, *PTEN* being a tumor suppressor gene [4,46,47].

Cowden syndrome 1 (CWS1) (OMIM 158350) [6] is a syndrome with multiple hamartomas associated with an increased risk for benign and malignant tumors of the thyroid, breast, kidney, and endometrium [6]. Clinical manifestations specific to CWS1 include macrocephaly, facial trichilemmomas, acral keratoses, and papillomatous papules [46]. The risk of developing breast cancer is 85% among CWS1 patients aged between 38 and 46 years, while the risk for follicular thyroid cancer (rarely papillary, but never medullary) is estimated at 35%, the risk of renal cell carcinoma (predominantly papillary) is 34%, and that of endometrial cancer is 28% of individuals with CWS1 [46,47]. In a prospective multicenter study, which included 2912 patients with CWS, from different geographic regions (North America, South America, Europe, Australia, and Asia), Ngeow et al. [48] showed that adult patients with PTEN mutations had a seven-fold increased risk of developing a second malignant neoplasm compared to individuals in the general population [48].

Bannayan-Riley-Ruvalcaba syndrome (BRRS) is a congenital condition characterized by macrocephaly, intestinal hamartomatous polyposis, lipomas, and pigmented macules of the glans penis [6,46].

PTEN-related Proteus syndrome (PS) is a complex disorder with variable phenotype, which associates congenital malformations and the development of hamartromas at the level of multiple tissues, as well as connective tissue nevi, epidermal nevi, and hyperostosis [46].

PTEN-related Proteus-like syndromes refers to individuals with characteristic phenotypic manifestations of PS, but who do not meet the diagnostic criteria for PS and who have a heterozygous germline *PTEN* pathogenic variant [46].

#### 3.2.2. Hereditary Hemorrhagic Telangiectasia (HHT)

Hereditary hemorrhagic telangiectasia (HHT) (Osler-Weber-Rendu syndrome) is a rare autosomal dominant genetic disorder, characterized by genetic heterogeneity, being determined by mutations of some genes located at the level of five loci (Table 1) [49]. The specific manifestations of HHT are multiple cutaneous lesions (telangiectasias) often associated with multiple AVMs that lack intermediate capillaries and result in direct connections between arteries and veins [49]. AVMs most commonly occur in the lungs, liver, and brain. The disease is manifested by recurrent epistaxis with onset before the age of 10, which frequently causes anemia, while cutaneous telangectasias (small AVMs) appear later in life and are located on the face, oral mucosa, and fingers [49]. Large AVMs are most often located in the lung, liver, or brain, causing sudden, massive bleeding and extremely severe complications. In some cases of HHT, affected individuals have gastrointestinal bleeding, but this is rarely seen before the age of 50 [49].

HHT is a disease characterized by genetic heterogeneity, caused by mutations of one of the multiple genes involved in the TGFB signaling pathway, which regulates cell proliferation, differentiation, apoptosis, and migration [9].

LOF mutations of the *ENG* gene (located on chromosome 9q34.11) (OMIM 18301) [6], which encodes endoglin, are present in HHT1 [6,50], and mutations of the *ACRVL1/ALK1* gene (OMIM 601284) (located on chromosome 12q13.13) [6] which encodes ALK1 (Activin A receptor, type II-like kinase 1 also called activin receptor-like kinase-1) are associated with HHT2 [51,52]. Other loci identified in patients with HHT are located on chromosome 5q31.3-32 (HHT3) [53] and chromosome 7p14 (HHT4) [54]. In some cases of HHT, mutations of the *GDF2* gene (growth/differentiation factor 2, also named Bone morphogenetic protein 9, BMP9) (OMIM 605120), located on chromosome 10q11.22 [6], which is the ligand for ALK1, have been detected [55]. Heterozygous LOF mutations of the *SMAD4* gene located on chromosome 18q21.2 [6] are associated with juvenile polyposis/hereditary hemorrhagic telangiectasia syndrome (JPHT) (OMIM 175050) [6]. The *SMAD4* gene encodes a protein (SMAD4) required for most transcriptional responses to TGFB (Transforming growth factor beta) signaling and BMs (bone morphogenic proteins, e.g., BMP1) [6,49,56,57].

HHT-related genes encode proteins that play a role in BMPs signaling. *ENG* and *ACVRL1* are expressed on ECs and encode TGFB pathway receptor proteins that are involved in the phosphorylation of SMAD proteins and the regulation of downstream signaling [6,49].

ALK1-mediated endothelial BMP9 and BMP10 signaling plays many important roles in angiogenesis and the maintenance of vascular quiescence. Endoglin induces BMP9/BMP10/ALK1 signaling by phosphorylating the receptor and activating the transcription factors R-SMAD (receptor-regulated SMAD) 1/5/8 and SMAD4, which normally suppress ECs migration and proliferation. LOF mutation of these genes causes increased migration, , and formation of blood vessels. VEGF and AKT signaling are also increased [52].

### 3.3. Lymphatic Malformations and PI3K/AKT/mTOR Signaling Pathways

According to the ISSVA Classification, 2018 [2] simple lymphatic anomalies are classified as Lymphatic malformations (LMs) and Primary lymphedema [2]. LMs include cystic LMs, generalized lymphatic anomaly (GLA) (which includes Kaposiform lymphangiomatosis, KLA), LMs in Gorham-Stout disease, “acquired” progressive lymphatic anomaly, and channel type LM. LMs can be isolated or syndromic (e.g., PROS) [2].

Lymphedema can be congenital or acquired, and consists of an accumulation of interstitial fluid that causes swelling of the affected limb. Primary lymphedema represents the “clinical manifestation” of lymphatic malformation as the result of defective development of the lymphatic system in its “later” stage of lymphangiogenesis. It was classified as the “truncular” type in the Hamburg Classification of Vascular Malformations. This classification also includes the “extratruncular” type, which represents the defective development of the lymphatic system in its “early” stage, which the ISSVA Classification called “lymphatic malformation”. Therefore, LM and primary lymphedema on the ISSVA Classification, are both two different types of the same lymphatic malformation originated from two different stages of embryogenesis [2,58].

The etiology of congenital lymphedema is extremely heterogeneous, with more than 20 mutant genes being described. Milroy’s disease, the classic form of congenital lymphedema, is transmitted in an autosomal dominant manner and is determined by LOF of the *VEGFR3* (vascular endothelial growth factor receptor 3 also called *FLT4*) gene, located on chromosome 5q35.3 [6]. Congenital lymphedema often presents at birth or childhood with swelling of both lower extremities [58,59].

Generalized lymphatic anomaly (GLA) is a severe form of diffuse or multifocal LMs. GLA is a rare, benign congenital disorder characterized by abnormal proliferation of lymphatic vessels resulting in dilated and abnormally connected thin-walled lymphatic channels. The disease manifests itself in children and young adults, the lungs and skeleton being most frequently affected, in the latter case causing pain and impaired mobility [59,60].

The etiology of LMs can be easily identified if LMs belong to a syndromic form of the disease (e.g., KTS, CLOVES or PROS). However, identifying the etiology of isolated genetic LMs remains a challenge for future research. In the last decade, progress has been made in identifying genetic factors involved in the development of LMs [60].

One of the genes of interest for the occurrence of LMs is the *PIK3CA* gene, which encodes the catalytic subunit of PI3K. The PI3K/AKT pathway is integral involved in lymphatic development by inducing the migration of lymphatic endothelial cells (LECs). The ASM mutations of the *PIK3CA* gene, especially in the p110α catalytic subunit, are detected in a large number of mixed vascular malformations [61]. In their study, Luks et al. [62] identified five activating mutations of the *PIK3CA* gene (p.C420R, p.E542K, p.E545K, p.H1047R, p.H1047L) that were present both in patients with isolated LMs and syndromic LMs (KTS, FAVA—Fibro-adipose vascular anomaly and CLOVES) [62]. The authors concluded that mutations in the *PIK3CA* gene are the most common cause of isolated LMs, and may also be detected in other diseases that associate LMs [62].

Most LMs have a hotspot mutation (p.Glu542Lys, p.Glu545Lys and p.Glu545Gly in the helical domain and p.His1047Arg and pHis1047Leu in the kinase domain) in the *PIK3CA* gene similar to those detected in VMs. Hotspot mutations in the *PIK3CA* gene, detected in LECs isolated from patients with LMs, were able to activate the PI3K/AKT/mTOR pathway. Heterozygous *PIK3CA* mutations are severe, being associated with an increased lethal risk. This could explain the absence of reporting of isolated inherited LMs, while new (de novo) non-hotspot mutations have been detected in some patients with syndromic LMs (MCAP) [4]. The type of mutation (de novo germline or postzygotic mutations) and the time of appearance of the postzygotic mutation (mosaic or somatic) are important. Compared to hotspot mutations, non-hotspot mutations are more often observed in mosaic and syndromic forms of LMs [63]. Moreover, the moment of activation of the *PIK3CA* H1047R mutation during murine development determined the LMs type. If *PIK3CA* H1047R was expressed early in fetal development in LECs, large macrocystic LMs developed, while postnatal expression of the mutation determined the appearance of microcystic LMs. These effects seem to be influenced by the activation of the vascular endothelial growth factor (VEGF)-C/vascular endothelial growth factor receptor (VGFR)-3 axis upstream of PI3K signaling [63]. VEGFR-3 is activated by its specific ligand VEGF-C, and promotes cancer progression. The VEGF-C/VEGFR-3 axis is expressed not only by LECs but also by a variety of human tumor cells [64]. Activation of the VEGF-C/VEGFR-3 axis in LECs can facilitate metastasis by increasing the formation of lymphatic vessels (lymphangiogenesis) within and around tumors. The VEGF-C/VEGFR-3 axis promotes cancer cell metastasis in some solid tumors and plays a critical role in leukaemic cell proliferation and resistance to chemotherapy [64].

## 4. Other Signaling Pathways

Apart from the PI3K/AKT/MTOR and RAS/RAF/ERK signaling pathways, other signaling pathways are also involved in VAs: VEGF-A/VEGFR2 signaling and PDGFB/PDGFRB Signaling Pathway in vascular tumors (IH) and HGF [hepatocyte growth factor])/c -Met signaling in glomuvenous malformations (GVMs) [4].

### 4.1. Vascular Tumors (Infantile Hemangioma): VEGF-A/VEGFR2 Signaling and PDGFB/PDGFRB Signaling Pathway

#### 4.1.1. VEGF-A/VEGFR2 Signaling Pathway

Infantile haemangioma (IH), also known as a strawberry naevus, is the most common benign vascular skin tumor in children. IH is present in 1–2% of Caucasian newborns and is usually seen in the first weeks of life [65].

The risk factors are female gender, low birth weight, prematurity, and maternal factors (advanced maternal age, infertility treatment, multiple pregnancy, pre-eclampsia, and placenta praevia). IH can have a variable distribution pattern (focal, segmental, multifocal, or indeterminate) [65]. The majority of IH presents a rapid proliferation phase in the first three months, with a growth arrest until the age of about 5 months, and then slowly involutes over time, until the complete regression of the tumor, which will be replaced by fibrous tissue and adipocytes [66].

IH with minimal or arrested growth (IH-MAG) has reduced growth in the form of telangiectatic spots with or without papules, lacking a significant proliferative phase, and can be confused with a CM (port-wine stain). Occasionally IH-MAG can have a segmental distribution (in 2/3 of the cases being located in the lower limbs), within a syndromic condition [65,66].

IH is produced by hyperplasia of ECs, but the underlying mechanism is not fully understood. Multiple theories (the clonality of ECs of the hemangioma and the theory of hypoxia and the successive angiogenesis) have been outlined, but the etiopathogenic mechanisms of IH are not yet fully elucidated [67].

To date, there are several hypotheses related to the etiopathogenesis of IH: (1) Placental hypothesis or theory of embolic placental angioblasts (expresses common placental markers, including GLUT1, Lewis Y antigen, Fcg and merosin); (2) The vasculogenesis theory or the embryonic endothelial precursor theory (circulating CD133+/CD34+ progenitor cells and stem cells are detectable in hemangiomas and in the blood of IH patients); (3) The hormonal theory: high levels of 17-beta-estradiol have been observed in several studies, simultaneously with a large number of estradiol receptors on the surface of hemangioma tissue in their proliferative phase; (4) The intervention of genetic factors (genes that intervene in the VEGF-A signaling pathway) [67,68].

Some patients with IH present a heterozygous germline missense mutation in the extracellular region of the *VEGFR2* gene which encodes vascular endothelial growth factor receptor 2 (VEGFR2), while other patients are presented a mutation in the *ANTXR1* (Anthrax toxin receptor 1, *TEM8*) gene (OMIM 606410), which affects the transmembrane domain of TEM8/ANTXR1 protein [67].

TEM8 is a tumor-specific endothelial marker highly expressed in tumor ECs but not in normal ECs [6]. The expression of the *TEM8* mutant gene in ECs suppresses β1 integrin (ITGB1) activity and represses the activation of NFAT (Nuclear factor of activated T cells) a transcription factor regulated by calcium influx, causing the decrease of VEGFR1 expression and the increase of VEGFR2 signaling and the proliferation of hemangioma ECs due to increased binding of VEGF to VEGFR2 [67,69,70].

Suppression of β1 integrin activation is a hallmark of all hemangioma ECs, even in cases where no mutations in *VEGFR2* or *TEM8* were found. It appears that suppressed β1 integrin activation in all ECs isolated from proliferating hemangioma lesions (hemECs) is associated with increased interactions between several components of the VEGFR2/TEM8/β1 integrin complex, which most likely also contains other components, which are not yet identified [67,68].

Several other factors are linked to the proliferating phase, including basic fibroblast growth factor (FGF), insulin-like growth factor (IGF-1), matrix metalloproteinase 9 (MMP-9), and the receptor tyrosine kinases TIE2 and TIE1 with the ligand angiopoietin-2 (ANGPT2). Carsten et al. [67] analyzed 185 patients with IH, one third of whom had a positive family history, suggesting two possible mechanisms of transmission of inherited IH: autosomal dominant with incomplete penetrance, or maternal transmission. In addition, the authors emphasized the need for additional studies to define inheritance of this common disease [67].

#### 4.1.2. PDGFB/PDGFRB Signaling Pathway

Walter at al. [70] provided the first evidence for a regulatory role of PDGFB/PDGFRB (platelet-derived growth factor receptor, beta) signaling in IH [70]. In three cases of familial IH, a link with the 5q locus was identified. The region 5q31-33 contains three candidate genes involved in blood vessel growth: fibroblast growth factor receptor 4 (*FGFR4*), *PDGFRB*, and *VEGFR-3* [70]. Subsequently, Calicchio et al. [71] in a study that aimed to identify differentially expressed genes in proliferating and involuting hemangiomas, showed a reduction in PDGFRB expression during the involuting phase of IH [71]. This finding suggests the possibility that PDGFB/PDGFRB signaling may play a role in the pathogenesis of IH [71].

### 4.2. HGF [Hepatocyte Growth Factor])/c-Met Signaling in Glomuvenous Malformations (GVM)

Glomuvenous malformation (GVM) (OMIM 138000) [6] represents a rare venous malformation, with variable number of mural glomus cells in the walls of distended venous channels. The prevalence of GMNs is not known exactly, but it is estimated that they represent approximately 70–80% of inherited VMs [72].

GVMs are characterized by the presence of some small, multifocal bluish-purple vascular cutaneous lesions mainly involving the skin and subcutaneous tissue (located especially at the extremities). GVMs are often painful on palpation and cannot be completely emptied by compression. They can cause paroxysmal pain, which occurs either spontaneously, through compression, or after trauma. In some cases, with extensive congenital GVM (especially in newborns), the lesions appear in the form of pink-purple plaques (which can be confused with a capillary malformation), whose color darkens with time. The disease is characterized by variable expressivity, and the phenotype can vary in affected family members from small flat lesions to a large plaque-like GVM-lesions (some family members may have only a few lesions, while others may have several hundreds).

Histologically, GVM is characterized by the presence of abnormally differentiated vascular smooth muscle cells (VSMCs), called glomus cells, which are located around dilated veins that are covered by abnormal, flattened ECs [72,73,74].

GVMs are caused by LOF mutations in the *GLMN* gene (located on chromosome 1p22.1) [6] encoding glomulin, a phosphorylated protein that is a member of a Skp1-Cullin-F-box-like complex which is essential for normal vascular development [6]. Inherited GVMs are transmitted in an autosomal dominant manner and cause the loss of glomulin function. The lesions develop in areas where a second -hit mutation has occurred, indicating that GVMs lesions are due to complete loss-of-function of glomulin. The most frequent somatic second-hit mutation of *GLM* gene is due to an acquired uniparental isodisomy, which leads to loss of the normal copy, and duplication of the mutated copy in cells within the lesion [72,73].

Brouillard et al. [74] analyzed 207 patients with GMVs, in 156 of them *GLMN* gene mutations were identified [74]. To date, 40 different mutations in the *GLMN* gene have been identified, which are present in 162 families with GVMs. Most mutations were detected in patients with a positive family history (143/162 cases; 88%), while only 19 cases were considered sporadic [74,75]. Glomulin interacts with the unphosphorylated c-Met HGF receptor. Binding of hepatocyte growth factor (HGF), a mediator of VSMC migration to its receptor induces phosphorylation of glomulin and its release. Subsequently, glomulin triggers the activation of kp70S6K kinase (a downstream target of PI3K). Glomulin appears to be involved in protein degradation by the ubiquitin-proteasome pathway by interaction with Cul7 (Cullin-7) which, in turn, forms a Skp1-Cul1-Fbox E3 ubiquitin-protein ligase complex [79].

Glomulin is specifically expressed at the VSMC level and seems to be involved in VSMC differentiation by interacting with the TGFB signaling pathway. Binding of FK506 binding protein 12 (FKBP12) to TGF-β type I receptor (TGFB-1 receptor) can inhibit TGFB signaling. In vitro, it has been observed that glomulin can interact with FKBP12, activating TGF-β. FKBP12-glomulin complex formation is inhibited by FK506 and Rapamycin leading to the hypothesis that glomulin may play a role in the mTOR signaling pathway [79].

Thus, taking into account the interaction of glomulin with the two signaling pathways (TGFB and mTOR), the modulators of the two signaling pathways could constitute possible targets for GVMs therapy [79].

## 5. Discussions and Future Therapeutic Perspectives Correlated with Molecular Mechanisms

Deciphering the genetic architecture of VAs (vascular malformations and tumors) present in the case of inherited genetic syndromes (germline mutations), as well as the identification of specific somatic mutations, detected in both syndromic and sporadic forms, played an important role in understanding the complex pathophysiological mechanisms of VAs, but also in deciphering the structure and normal function of blood vessels.

Starting from the abnormal structure of the vascular malformations, it was concluded that they are caused by mutations of the genes involved both in the development and function of ECs, VSMCs, and proteins in the extracellular matrix [4]. In addition, these genes encode numerous molecules that intervene in different signaling pathways that can alter the normal function of ECs and VSMCs. Germline mutations have been identified in some of the inherited VAs, and since 2002, the second-hit mutation mechanism was identified, which causes, in most cases, the LOF of the protein encoded by the involved gene [1,4].

In most isolated VMs, which appear sporadically, somatic GOF mutations are frequently detected, and the first essential discovery was the identification in 2009 of somatic mutations of the *TIE2* gene (angiopoietin-1 receptor) in VMs [4,5]. Many of the identified mutations are located in genes that play an important role in signaling pathways involved in angiogenesis and lymphangiogenesis, vascular cell growth, cell proliferation, and apoptosis. Another aspect that must be taken into account is related to the fact that many of these mutations are detected in different types of cancer. Starting from this aspect, the effectiveness of anticancer drugs in the treatment of vascular anomalies was discussed.

The main signaling pathways involved are angiopoietin/TIE2/PI3K/AKT/mTOR), RAS/RAF/MAPK/ERK, and the TGFB signaling pathway. G protein-coupled receptor signaling molecules (GNAQ/GNA11/GNA14) may also be involved. The TGFB signaling pathway and PI3K were already known to be involved in cancer biology, but despite all this, their role in vascular biology and the pathophysiology of vascular disease was either unknown or underappreciated [4].

The elucidation of the etiology of VAs still remains a very current topic and constitutes a challenge both for clinicians and for future research studies. Phenotypic variability correlated with genetic heterogeneity, as well as phenotypic overlaps between different entities, can create difficulties in terms of clinical diagnosis [4].

Considering these aspects, in the future, the approach to patients with VAs could also include genetic testing, along with radiological diagnosis and interventional surgery [4]. Multidisciplinary management and cooperation between geneticists and clinicians will most likely lead to a decrease in the mortality and morbidity of VAs [4,5].

An eloquent example is the genetic testing of *RASA1* gene mutations, which helps to differentiate the diagnosis of patients with port-wine stain type CMs (PWS, KTWS and CM-AVMs) [4,16,18].

The development of specific genetic tests would lead to improve the diagnosis in the syndromic forms allowing the initiation of a specific targeted therapy [1,5].

### 5.1. Current Approaches and Future Challenges in The Treatment of Vascular Anomalies: New Molecular Targeted Therapies

The initial treatment, limited to large and extensive vascular malformations, represented by sclerotherapy, embolization, surgical intervention, or laser ablation is the gold standard in the management of VAs. These procedures are rarely curative, having a high risk of recurrence and are associated with increased morbidity. In these cases, the patients present severe chronic pain, and sometimes, significant destruction of tissues, affecting the quality of life. With the deciphering of the pathophysiological mechanisms of VAs and the signaling pathways involved, a new stage of research has begun, regarding the use of anticancer therapy in the treatment of VAs [4,5].

The discovery of new genes (e.g., encoding RNF213 and glomulin) and the subsequent deciphering of their function will open new perspectives regarding the etiopathogenesis of vascular pathology and could be the basis for the development of new personalized therapies that act on a target specific [4,5].

The proteins encoded by the *CCM* genes, identified in the case of cerebral cavernous malformations (CCMs), seem to modulate the activity of several signaling pathways that interact with each other. The biochemical connections between these suggest that certain therapies may be effective in multiple types of VMs. The study of abnormal proteins could facilitate the understanding of the molecular mechanisms present in the case of VMs [8,10,11].

Rapamycin (also called Sirolimus or Rapamune) is a specific inhibitor of mTOR preventing the phosphorylation of Ribosomal protein S6 (S6RP) and 4E-BP1 (eukaryotic translation initiation factor 4E-binding protein 1), by mTORC1 (Mammalian target of rapamycin complex1) [4,5]. The role of Rapamycin in VAs was highlighted relatively recently, being initially used in clinical practice as an immunosuppressive, antiangiogenic, and cytostatic agent [4].

Mammalian target of Rapamycin (mTOR) is a protein kinase that controls cell growth, proliferation, and survival. The mTOR signaling pathway is often overregulated in cancer and there is great interest in the development of selective mTOR kinase inhibitors as a new class of anticancer drugs [4,80,81].

In some preclinical studies, treatment with Rapamicyn or Everolimus confirmed its effectiveness in delaying the growth and reducing the volume of slow-flow VMs. Rapamycin can thus block progression of the vascular injury and restores the vascularization of the respective tissue, the process being mediated by the disruption of mTORC2 (Mammalian target of rapamycin complex 2) by Rapamycin and its impossibility to phosphorylate Ser473 residue of AKT. Reduced activation of AKT will cause increased levels of active transcription factor FOXO1 and increased levels of pericyte attractant PDGFB, which may help reduce ECs proliferation [4,80,81].

Various retrospective clinical studies have revealed the efficacy of Rapamycin in adults and children with complex, extremely severe Vas, such as kaposiform hemangioendothelioma, Kasabach-Merritt phenomenon, KLA (kaposiform lymphangiomatosis), capillary-lymphatico-VMs, and LMs. The initial dose of Rapamycin was 2 mg daily for adults and 0.8 mg/m^2^ [4].

The VASE study (Multicenter Phase III Study Evaluating the Efficacy and Safety of Sirolimus in Vascular Anomalies Refractory to Standard Care) is the largest prospective multicenter Phase III trial currently underway in patients with refractory slow-flow complex VMs to standard treatment [80].

The preliminary results in the case of 101 patients (70 cases VMs, 4 CMs, 14 cases LMs, 6 cases with KTS/CLOVES, 4 cases GLA, 1 case PHTS, 2 cases GSD) who were followed for a period of 6 months of treatment with Rapamycin showed that in over 87% of cases the quality of life improved, and pain and functional impotence decreased [80].

Studies done in cases of AVMs treated with Rapamycin showed modest or transient efficacy compared to slow-flow VMs, suggesting a reduced involvement of the PI3K/AKT/mTOR signaling pathway in the production of AVMs [4,81,82].

In a prospective study on 39 patients with PIK3CA-related overgrowth spectrum (PROS), the efficacy of Sirolimus treatment had low efficacy, without an increase in quality of life. In these patients, targeted inhibition of PI3K represents a promising future therapy [83]. In a prospective trial that included 19 patients with PROS (who had severe or life-threatening complications and were unresponsive to any other treatment) treated with Alpelisib (BYL719), a subunit-specific inhibitor p110α of PI3K, Venot et al. [84] noted an improvement in symptoms in 27% of cases [84]. In these patients, significant anatomical and functional improvements occurred, regardless of location and affected organs [84]. Pagliazzi et al. [85] showed efficacy of Alpelisib treatment in a patient with CLOVES [85]. After 12 months of treatment the authors observed a regression of adipose overgrowth and low-flow vascular malformation, with a decrease in basal D-dimer levels, without adverse effects. The authors concluded that Alpelisib is a promising targeted treatment in patients with overgrowth syndromes, with or without vascular malformations, caused by PI3K hyperactivity [85].

Mirarsertib is a selective AKT inhibitor that is currently being investigated in several types of cancer. The administration of Mirarsertib to a patient with Proteus syndrome and ovarian carcinoma, determined the regression of symptoms and the improvement of the quality of life [86]. Forde et al. [87] showed that Mirarsetib temporarily improved symptoms in a patient with CLOVES, as well as in a patient with facial infiltrating lipomatosis and hemimegalencephaly [87].

Trametinib is an inhibitor of the kinase activity of MEK1 and MEK2, and seems to be a potential treatment for VAs involving the RAS/RAF/MEK kinase pathway. These include some complex lymphatic anomalies (LAs) [4].

In the case of a patient with KLA/generalized lymphatic abnormalities (in whom the ASM mutation in *NRAS* was identified in lymphatic ECs), in vitro, Trametinib and Rapamycin reduced the viability of lymphatic cells. In a patient with central conducting lymphatic anomaly with activating *ARAF* (A-Raf proto-oncogene, serine/threonine kinase) mutation, Trametinib induced almost complete remodeling of the lymphatic system and remission of symptoms [88,89]. Trametinib seems to be a promising treatment also in the case of patients with AVMs [23]. Recently, the ASM mutation in the *KRAS* gene was detected in 45 of 72 patients with cerebral AVMs. This mutation induced increased ERK activity in AVM-derived ECs, enhancing angiogenesis and migratory behavior. These processes were reversed by inhibiting MAPK/ERK signaling [23].

A prospective phase II study, TAMAV (Evaluating the safety and efficacy of Trametinib in Arteriovenous Malformations that are refractory to standard care), is currently underway, which evaluates the effectiveness of Trametinib treatment in patients with AVMs refractory to standard care [90].

Thalidomide is a potent immunosuppressive and antiangiogenic agent effective in the treatment of inflammatory diseases and various types of cancer [4]. Thalidomide reduces nose bleeding in patients with HHT. Using a mouse model Zu et al. [91] showed that Thalidomide and Lenalidomide reduce brain AVM hemorrhage in which is likely through upregulation of PDGFB expression [91]. In a mouse model, Kim et al. [92] suggested that ALK1 overexpression can prevent AVMs formation in ENG-deficient mice and could be an effective therapeutic strategy for HHT. But further research is needed to see if this therapy could also be applied to humans [92].

Bevacizumab is an anti-VEGF monoclonal antibody designed to inhibit tumor induced neo-angiogenesis. Bevacizumab prevents the binding of VEGF to its receptors (VEGFR) inducing strong antiangiogenic effects [4]. Several studies have shown that treatment with Bevacizumab has favorable effects in the treatment of recurrent epistaxis and pulmonary, hepatic, and intestinal AVMs in patients with HHT, but data on its effectiveness in the case of isolated AVMs are not known [93,94]. A randomized phase III clinical trial to study the efficacy of Bevacizumab in patients with HHT who have severe bleeding requiring blood transfusions is ongoing [95].

Although we focused on an in-depth analysis of the data from the specialized literature regarding the genetic factors that encode components of the signaling pathways involved in the occurrence of VAs (vascular malformations and tumors), we still consider that our study was limited by incomplete data related to the etiology of VAs. On the other hand, the large number of entities, the phenotypic variability correlated with the genetic heterogeneity, and the phenotypic overlaps between different VAs can create difficulties related to diagnosis and the initiation of an early treatment. Deciphering the etiology of VAs is a fascinating topic that remains relevant due to the complexity of the possible factors involved and the interaction between them.

In addition, future research will most likely identify new therapeutic targets, with the possibility of developing modern therapies, which will improve the prognosis of patients, especially in severe forms of the VAs.

### 5.2. Genetic Counseling in Vascular Anomalies: Inherited Versus Sporadic Vascular Anomalies

VAs can be caused by inherited germline mutations or acquired somatic mutations. In most VAs that occur sporadically, postzygotic somatic mutations are detected. In their case, a somatic mosaicism is present, and the genetic testing is usually performed from the affected tissue obtained by surgical or skin biopsy [76,96]. In the case of germline mutations (present in all body cells in affected individuals), testing can be done from any type of tissue (e.g., skin, blood, hair, or saliva) [76].

Although rarer than sporadic forms, familial or inherited vascular malformations or syndromes provide unique insight into the molecular mechanisms that control vascular morphogenesis [76]. Most germline mutations are transmitted in an autosomal dominant manner with incomplete penetrance and determine the LOF of the affected gene. In some cases, the onset is late, suggesting the need for the production of the second-hit mutation for the appearance of the tumor or vascular malformations [76].

In addition, the inherited VAs are characterized by variable expressivity in terms of clinical manifestations and localization of the anomalies [97]. Family anamnesis and analysis of the family tree may suggest that the disease is inherited, being determined by a germline mutation. If the germline mutation present in the family is already known, in the event of a new pregnancy, molecular testing and prenatal diagnosis are possible. For example, the identification of the mutation and the confirmation of the diagnosis of PHTS requires an early management with periodic monitoring, taking into account the increased risk of childhood-onset thyroid cancer and the appearance of other malignant tumors in adulthood [76].

It is known that LOF mutations are often private and unique to the affected individual or family and do not cluster into hotspots, while activating GOF mutations tend to occur at hotspots. Some new mutations may be reported as “variants of unknown significance”, requiring careful phenotype characterization to confirm the link between the identified genetic change and clinical manifestations. Many of these show an autosomal dominant inheritance of the disease, with the first mutation being inherited via germline cells and a second mutation acquired in the somatic cells (according to Knudson’s two-hit hypothesis) [98]. In current practice, in cases where constitutional variants (germline mutation) are identified, the genetic counseling will necessarily include the complete family history, in order to assess the risk of an inherited condition. The detection of constitutional variants is usually carried out by gene sequencing, with the possibility of excluding low variant allele frequency (VAF) that is somatic as a possible sequencing error. A variant with a high VAF could also be confused with a constitutional variant [98].

The identification of genes involved in VAs is essential for deciphering the complex genetic architecture of their etiology as well as for new targeted therapeutic approaches. As the specific molecular changes will be identified, there will probably be a redefinition of certain phenotypes, and an improvement of the clinical classification, as well as the possibility of a phenotype-genotype correlation. Genetic testing and molecular diagnosis will create the possibility of developing new, personalized therapies, based on the specific molecular mechanism [76].

## 6. Conclusions

In recent years, important advances have been made in elucidating the complex etiology of VAs with the help of increasingly advanced molecular technologies. The main benefit was, on the one hand, the correct diagnosis, essential for recommending the optimal treatment, and, on the other hand, the identification of new therapeutic targets and the development of modern therapies that will determine the improvement of the prognosis and life expectancy, especially in the severe forms of VAs.

Understanding the genetic and molecular basis of VAs is essential for clinicians, as this information will likely cause the redefinition of some phenotypes correlated with genetic etiology, a new classification, and will provide new treatment options in the future.

Modern therapies in VAs target the signaling pathways involved (RAS/MEK/ERK and/or PI3kinase/AKT/mTOR), and their development will only be possible by identifying the genes and proteins encoded by them, as well as the consequences of gene mutations in the signaling pathways.

Frequently, these mutations influence the proliferation, differentiation, and survival of altered ECs. In addition to sclerotherapy and surgical excision, protein inhibitors involved in the signaling pathways PI3K/AKT/mTOR or RAS/BRAF/MAPK/ERK appear as targeted therapeutic options for different types of vascular malformations, so far, the clinical results being promising. The discovery that many of the genes involved in VAs are also involved in different types of cancer was the basis for a new therapeutic approach based on the use of anticancer drugs in the therapy of VAs.

Geneticists play an essential role in the multidisciplinary team to be involved in the management of VAs, both in confirming the diagnosis and providing genetic counseling in cases of VAs. In late-onset inherited cases, providing genetic counseling can be difficult, being suggested the need to produce the second-hit mutation for the appearance of tumor or vascular malformations. In addition, the second-hit mechanism could explain the localized nature, multifocality, variable expressivity, and incomplete penetrance of these lesions that reach their maximum at puberty.

## Figures and Tables

**Figure 1 ijms-23-12199-f001:**
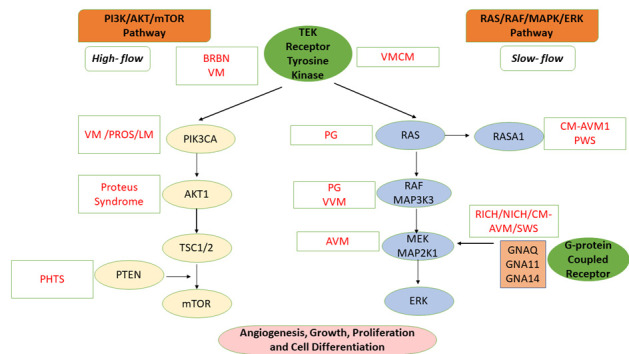
Molecular Mechanism of Vascular Anomalies: PI3K/AKT/mTOR and RAS/MAPK/ERK Signaling Pathways [1,4,5]. VM: Venous malformation; LM: Lymphatic malformation; AVM: Arteriovenous malformation; BRBN: Blue rubber bleb nevus (BRBN) syndrome; PHTS: PTEN hamartoma tumor syndrome; CM-AVM: Capillary malformation—Arteriovenous malformation syndrome; PROS: PIK3CA-related overgrowth spectrum; RICH: Rapidly involuting congenital hemangioma; NICH: Noninvoluting congenital hemangioma; VMCM: Cutaneomucosal venous malformation; PWS: Parkes Weber syndrome; VVM: Verrucous venous malformation; PG: Pyogenic granuloma; SWS: Sturge-Weber syndrome; *MAP2K1*: Mitogen-activated protein kinase kinase 1; ERK: Extracellular signal-regulated kinase; mTOR: Mammalian target of Rapamycin; PTEN: Phosphatase and tensin homolog; GNAQ: Guanine Nucleotide-Binding Protein G(Q) Subunit Alpha; TSC1/2: Tuberous sclerosis complex 1/2; TEK: TEK tyrosine kinase; RAF: Rapidly accelerated fibrosarcoma; PIK3CA: Phosphatidylinositol 4,5-Bisphosphate 3-Kinase Catalytic Subunit Alpha; RASA1: RAS p21 protein activator 1.

**Table 1 ijms-23-12199-t001:** Classification of Vascular Anomalies According to ISSVA (2018) [2].

Vascular Anomalies	
*Vascular Tumors*	Benign
	Locally aggressive or Borderline
	Malignant
*Vascular Malformations*	
Simple	CM/VM/LM/AVM */AVF *
Combined	Defined as ≥2 vascular malformations found in one lesion
Anomalies of major named vessels (“channel type” or “truncal” vascular malformations)	Abnormalities in the origin/course/number/lenght/diameter (aplasia, hypoplasia, stenosis, ectasia/aneurysm)/persistence (of embrional vessels)/communication (AVF) of major blood vessels that have anatomical names
Associated with other anomalies	Syndromes that associate vascular malformations with non-vascular symptoms

ISSVA: International Society for the Study of Vascular Anomalies; CM: Capillary malformation VM: Venous malformation; LM: Lymphatic malformation; AVM: Arteriovenous malformation; AVF: Arteriovenous fistula; * high-flow lesions.

**Table 2 ijms-23-12199-t002:** The Molecular Pathophysiology of Vascular Anomalies: Genes/Loci and Signaling Pathways.

Vascular Anomalies	OMIM	Gene	Locus	Type of Mutation	Other Involved Pathway(s)	Reference
**RAS/RAF/MEK/ERK Signaling**						
*Venous anomalies*						
VVM	602539	*MAP3K3*	17q23.3	ASM		[6,7]
CCM	116860	*KRIT1*	7q21.2	LOF	ß-integrin signaling?	[8,9,10]
	603284	*CCM2 (Malcavernin)*	7q13	LOF	Notch signaling	[11]
	603285	*PDCD10*	3q26.1	LOF	Notch signaling	[12]
	619538	*MAP3K3*	3q26.3-27.2	LOF	Notch signaling	[13]
*Capillary anomalies*						
CM/SWS	185300	*GNAQ*	9q21.2	ASM		[14,15]
CM-AVM1	608354	*RASA1*	5q14.3	LOF	FAK signaling	[16,17,18,19]
CM-AVM2	618196	*EPHB4*	7q22.1	LOF		[20]
*Arteriovenous anomalies*						
Sporadic extracranial AVM	176872	*MAP2K1*	15q22.31	ASM		[21,22,23]
	190070	*KRAS*	12p12.1			[24]
	164757	*BRAF*	7q34			[24]
Brain AVM	108010	*KRAS*	12p12.1	ASM		[23]
*Lymphatic anomalies*						
GSD	123880	*KRAS*	12p12.1	ASM		[25,26]
KLA	164790	*NRAS*	1p13.2	ASM		[27,28]
*Vascular tumors*						
PG	163000	*GNAQ (secondary PG)*	9q21.2	ASM		[29]
	190070	*KRAS*	12p12.1	ASM		[29]
	190020	*HRAS*	11p15.5			[30,31]
	164757	*BRAF* *(isolated PG)*	7q34	SHM		[29,32]
	604397	*GNA14*	9q21.2	SHM		[27]
Congenital hemangioma (RICH, NICH)	600998, 139313	*GNAQ, GNA11*	9q21.2, 19p13.3	ASM	YAP signaling	[33,34]
**PI3K/AKT/mTOR Signaling**						
*Venous anomalies*						
Sporadic VM	600221	*TEK (L914F)*	9p21.2	ASM	RAS/MAPK/ERK signaling?	[35,36]
VMCM	600195	*TEK* (R849W)	9p21.2	ASM	RAS/MAPK/ERK signaling?	[37,38]
MVM	600221	*TEK*	9p21.2	ASM	RAS/MAPK/ERK signaling?	[39]
BRBN	112200	*TEK*	9p21.2	ASM		[39,40]
*Venous anomalies and PIK3CA-related overgrowth syndrome*						
MCAP	602501	*PIK3CA*	3q26.32	ASM		[41,42,43]
CLOVES	612918	*PIK3CA*	3q26.32	ASM		[44,45]
*Arteriovenous anomalies*						
PHTS	601728	*PTEN*	10q23.31	LOF		[46,47,48]
HHT	187300	*ENG*	9q34.11	LOF	BMP9/10/ALK signaling	[49,50]
	601284	*ACVRL1 (ALK1)*	12q13.13	LOF		[51,52]
	601101	*HHT3*	5q31.3-32	LOF		[53]
	610655	*HHT4*	7p14	LOF		[54]
	605120	*GDF2 (BMP9)*	10q11.22	LOF		[55]
JPS/HHT (JPHT)	175050	*SMAD4*	18q21.2	LOF		[49,56,57]
*Lymphatic anomalies*						
LM	613089	*PIK3CA*	3q26.32	ASM		[58,59,60,61,62,63,64]
**Other Signaling Pathways**						
*Vascular tumors*						
IH	191306	*VEGFR2*	4q12	Variants	Increased VEGFR2 signaling; reduced VEGFR1 signaling	[65]
	606410	*ANTXR1 (TEM8)*	2p13.3			[66,67,68,69]
		*VEGFR3?* *PDGFRB?* *FGFR4?*	5q31-33			[70,71]
*Venous anomalies*						
GVM	138000	*GLMN*	1p22.1	LOF	HGF/c-Met signaling; TGFB signaling?	[72,73,74,75]

LOF mutations: Loss-of-function mutation; ASM: Activating somatic mutations: SWS: Sturge-Weber syndrome; SHM: Second-hit mutation; IH: Infantile hemangioma; AVM: Ateriovenous malformation; BRBN: Blue rubber bleb nevus syndrome; CCM: Cerebral cavernous malformation; CLOVES: Congenital lipomatous overgrowth with vascular anomalies, epidermal nevi and scoliosis; CM: capillary malformation; CM-AVM1: Capillary malformation-arteriovenous malformation 1; GSD: Gorham-Stout disease (Cystic angiomatosis of bone, diffuse); GVM: Glomuvenous malformation; HCCVM: Hyperkeratotic cutaneous capillary-venous malformation; HHT: Hereditary hemorrhagic telangiectasia; KLA: Kaposiform lymphangiomatosis; LM: Lymphatic malformation; MCAP: Megalencephaly–capillary malformation; MVM: Multifocal venous malformation; PG: Pyogenic granuloma; PHTS: PTEN hamartoma tumor syndrome; RICH: Rapidly involuting congenital hemangioma; NICH: Noninvoluting congenital hemangioma; VM: Venous malformation; VMCM: Cutaneomucosal venous malformation; VVM: Verrucous venous malformation.

## Data Availability

Not applicable.

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
