# Peer review of "The Genetic Architecture of Vascular Anomalies: Current Data and Future Therapeutic Perspectives Correlated with Molecular Mechanisms"

_ijms, 2022, doi:10.3390/ijms232012199_

Round 1

Reviewer 1 Report

Dear Authors

Excellent review I enjoyed very much as a clinician involved to the care of vascular malformations for decades. But certainly I feel further needs to verify the ISSVA Classification of Vascular Anomaly (VA) more clearly since it accommodates two different vascular issues involved to the Vascular Malformation and Vascular Tumor, as suggested through the recommendation. 

1. Line 49-50 (Page 2) Introduction: Vascular anomalies (VAs) deserved for further explanation  to clarify such critical differences between Vascular Tumor and Vascular Malformation. It should be described clearly to avoid the confusion.

2. Line 62. Table 1: Change/correct ‘Malign’ to 'Malignant'.

3. Line 69: Change/correct ‘develope’ to 'develop'.

4. Line 164: Instead of 'presence', I recommend 'cluster' with further delineation of these embryonic tissue remnants for its capacity to maintain/possess the mesenchymal cell characteristics as the result of developmental arrest/error in its early stage of embryogenesis !

5. Line 442: I prefer 'Vascular Malformation' to replace 'VA' since there is NO 'slow-flow' Vascular Tumor.

6. Line 624: Current statement “ Lymphedema can be congenital” is NOT sufficient to avoid unnecessary confusion! It deserved more clarification as following; “Primary lymphedema represents 'clinical manifestation' of lymphatic malformation as the result of defective development of lymphatic system in its 'later' stage of lymphangiogenesis. It was classified to "truncular" type on Hamburg Classification of vascular malformation together with "extratruncular" type representing the defective development in its 'early' stage which ISSVA Classification named to 'lymphatic malformation'. Hence, LM and Primary Lymphedema on ISSVA Classification, BOTH are two different types of same lymphatic malformations originated from two different stages!!! 

All the best,

A Reviewer

Author Response

Dear reviewer,

We appreciate you taking the time to review our article. Your suggestions were considered, and the necessary adjustments were made.

  1. Line 49-50 (Page 2) Introduction: I reformulated and explained the differences between vascular tumors and vascular malformations as you suggested.
  2. Line 62. Table 1: Change/correct ‘Malign’ to 'Malignant: I changed.
  3. Line 69: Change/correct ‘develope’ to 'develop': I changed.
  4. Line 164: Instead of 'presence', I recommend 'cluster': I changed.
  5. Line 442: I prefer 'Vascular Malformation' to replace 'VA': I changed.
  6. Line 624: I reformulated the paragraph as you suggested.

Thank you!

Reviewer 2 Report

Comprehensive analysis well conducted. Authors sustain that Modern therapies in VAs target the signaling pathways involved (RAS/MEK/ERK and/or PI3kinase/AKT/mTOR), and their development will only be possible by identifying the genes and proteins encoded by them, as well as the consequences of gene mutations in the signaling pathways. They also confirm that geneticists play an essential role in the multidisciplinary team to be involved in the management of VAs, both in confirming the diagnosis and providing genetic counseling in cases of VAs.

Author Response

Dear reviewer,

We appreciate you taking the time to review our article. 

Thank you!